# Impact-Type Sunflower Yield Sensor Signal Denoising Method Based on CEEMD-WTD

**Shuai Wang** [1], **Xiaodong Zhao** [2], **Wenhang Liu** [1], **Jianqiang Du** [1], **Dongxu Zhao** [1] **and Zhihong Yu** [1,*]

[1] College of Mechanical and Electrical Engineering, Inner Mongolia Agricultural University, Hohhot 010018, China
[2] Inner Mongolia Electric Power (Group) Co., Ltd. Xilingol Power Supply Company, Xilinhot 026000, China
* Correspondence: yzhyq@imau.edu.cn

**Abstract:** During the crop harvesting process, it is important to obtain the crop yield quickly, accurately and in real time to accelerate the development of smart agriculture. This paper investigated a denoising method applicable to the impact-type sunflower yield sensor signal under the influence of complex noise background in the pneumatic seed delivery structure for a sunflower combine harvester. A signal processing method combining complementary ensemble empirical mode decomposition (CEEMD) and wavelet threshold denoising (WTD) based on an adaptive decomposition capability was proposed by analyzing the non-smoothness of the signal with the impact-type sunflower yield sensor signal in sunflower fields. CEEMD was used to decompose the sunflower seed impact analog signal and field impact-type sunflower yield sensor signal adaptively, and the high frequency components were processed by WTD. Finally the de-noised signal was obtained by reconstruction. An evaluation objective function of the denoising ability of the algorithm based on signal-noise ratio, root mean square error, smoothness and waveform similarity indexes with different weights was also constructed. The results showed that the evaluation objective functions of the simulated and measured signals after denoising by the CEEMD-WTD method are 1.9719 and 4.5318, respectively, which are better than the single denoising methods of EMD (1.5096 and 4.0012), EEMD (1.8248 and 4.0724), CEEMD (1.9516 and 4.3384), and WTD (1.8737 and 4.5294). This method provides a new idea for signal denoising of the impact-type sunflower yield sensor installed in the pneumatic seed delivery structure, and further provides theoretical support and technical references for the development of sunflower high-precision yield measurements in smart agriculture.

**Keywords:** smart agriculture; sunflower; pneumatic conveying; impact signal; CEEMD; WTD





## 1. Introduction

Sunflower (*Helianthus annuus* L.), an important agricultural cash crop and oil crop in China, is mainly grown in Inner Mongolia, Xinjiang, Jilin, and Gansu [1]. With the development of information technology, artificial intelligence and biotechnology, Chinese agriculture has gradually entered into a new stage of modern agriculture with digitalization, refinement, standardization and scale production, opening up an agriculture 4.0 model that is integrated with the Internet and highly intelligent [2]. It will face many challenges in the future [3,4], and will also obtain good development opportunities [5–7]. In the process of crop planting and production management, acquiring information on crop yield distribution in farmland in an online, real-time, and effective way is the main starting point for implementing precision agriculture [8,9], and is the primary task for improving the precision agriculture production management system and promoting smart agriculture. Along with the development and application of agricultural mechanization, sensor intelligence and Internet of Things technology, crop yield online monitoring technology has gradually developed into a key technical means of precision agricultural production management systems. Crop online yield measurement technology is an important guarantee to promote the intelligent development of agricultural machinery and equipment by

means of yield measurement sensors, moisture sensors and positioning devices installed in the combine harvester to achieve online real-time access to information on crop yield and its spatial distribution variability during the harvesting process. At this stage, domestic and foreign methods for obtaining crop yield information mainly include dynamic weighing measurement, volume measurement, impact measurement, and radiometric methods. The main applications of sensors include load cells with cores, such as resistance strain gauges and pressure ceramics, photoelectric sensors, image sensors, and radiation sensors with cores, such as X-ray and gamma-ray sources. Among them, impact force measurement methods are more widely used, and among these, mainly impact-type grain yield sensors are used [10]. This sensor consists of an impact plate and a force sensor, which converts the impact force averaged per unit time into a voltage signal. This sensor has a simpler structure; however, the measurement accuracy is susceptible to the vibration of the combine harvester.

To improve the yield measurement accuracy of the impact-type grain yield sensor, Zhou et al. [11] used a thin steel plate at the free end of the cantilever beam for the impact-type grain yield sensor to form a square cylindrical thin-walled member concentric with the cantilever beam, and used a polymer damping material to wrap the cantilever beam to eliminate the effect of vibration. Zhou et al. [12] and Hu et al. [13] designed a two-plate differential impulse grain flow sensor. One of the force sensors acts as a measuring beam and receives the impact force influence of grain flow directly, and the other acts as a reference beam, which is not influenced by the grain flow but only excited by the vibration of the sensor frame and its own mass, and, finally, the influence of body vibration on the accuracy of yield measurement is eliminated by a differential vibration elimination circuit. For the vibration noise of the impact-type grain yield sensor, Maertens et al. [14] designed a double-trap filter to remove the vibration noise by measuring the mechanical vibration and the vibration frequency of the lift carrier. Shoji et al. [15] reduced the singular value error by setting the output threshold of the impact plate at the maximum and minimum flow rate, and reduced the cumulative error by the sensor output zero. Qian et al. [16] used a low-pass filter, the addition of vibration isolators, and the design of a two-plate differential impact-type grain yield sensor to eliminate vibration disturbance signals. Cong et al. [17] applied a grain-shock frequency harmonic extraction method and a two-parallel beam signal adaptive interference pair cancellation method for vibration noise cancellation. Wei et al. [18] used a frequency domain signal processing method. Wang [19] conducted a FIR low-pass filtering method, which was performed on two parallel beam signals of the sensor. Chen et al. [20] proposed a de-noise processing method for the output signal of an impact-type grain yield sensor using wavelet transform. Li et al. [21] reduced the interference of mechanical vibrations by designing a curved impact plate and double plate differential, and used a double-threshold filtering method to improve the measurement accuracy.

Comprehensively, and beyond domestic and international development statuses, existing research scholars have mainly improved the measurement accuracy of impact-type grain yield sensors through sensor structure optimization and output signal denoising methods. In terms of structure, the addition of damping materials and additional vibration detection plates have mainly been used. The method is more costly, with a more complex design and installation process, and poor feasibility. The methods of sensor output signal denoising mainly include the filtering method, Fourier spectrum analysis method, wavelet transform method, etc. The denoising method can reduce the noise of the sensor output signal to varying degrees and improve the accuracy of measurement yields. The above denoising algorithms are only applicable to fixed frequency noise in the signal, and the measurement results are affected by different harvesting machinery and field operation environments. At the same time, when distinguishing the useful yield measurement signal from the useless noise, the key parameters need to be set artificially, and the inappropriate parameter settings will not be able to better distinguish the signal from the noise, and the existing research has not yet explored the research on the adaptive decomposition of the output signal of the impulse-type yield measurement sensor, which seriously limits the

further improvement of the yield measurement accuracy. At present, the impact-type yield sensor is mainly applied to the combine harvester whose net grain conveying structure is a scraper-type elevator for the yield monitoring of wheat, rice, and other grains. Compared with grains, sunflower seeds have huge differences in quality and form, and the net grain conveying of a sunflower combine harvester is a pneumatic grain conveying structure, and no research has been conducted on the signal denoising of the impact-type sunflower yield sensor. Therefore, this impact force yield measurement signal is different from that of grains, and cannot be directly applied or borrowed from the denoising algorithm of the impact force yield measurement signal of the scraper-type grain combine harvester.

Based on the above, this study explored the characteristics of the sensor output signal based on an impulse sunflower seed flow sensor applied to net grain conveying as a pneumatic structure, and conducted the following studies:

(1) A wavelet threshold denoising method based on complementary ensemble empirical mode decomposition (CEEMD) was proposed to denoise the sensor signal;

(2) The CEEMD-WTD method was compared with the single empirical mode decomposition (EMD), ensemble empirical mode decomposition (EEMD), and the field measured impulse sunflower seed flow sensor signals using numerical simulation signals and field measurements, including decomposition (EEMD), CEEMD, and wavelet threshold denoising (WTD) methods;

(3) The quantitative analysis of the denoising ability was carried out by the constructed algorithm denoising evaluation objective function;

The following problems can be solved.

(1) The shortcomings of two single algorithms, CEEMD algorithm and WTD algorithm, can be solved, and the fused CEEMD-WTD algorithm has adaptive decomposition ability and fast computation ability to meet the needs of random signal denoising and real-time yield measurement in field yield measurements;

(2) The shortcomings of the existing single EMD, EEMD, CEEMD and WTD methods can be solved, which have low applicability in terms of SNR, RMSE, NCC and f, and provide a feasible idea for the denoising of impulse crop flow sensor signals applied to pneumatic grain delivery structures.

## 2. Materials and Methods

### 2.1. Impact-Type Sunflower Yield Sensor

The grain combine harvester is a type of agricultural equipment that integrates multiple subsystems, such as walking, cutting, threshing, cleaning, and grain collection [22]. The 4ZXRKS-4 type self-propelled sunflower combine harvester is powered by a diesel engine and mainly consists of a cutting table, winch, over-bridge, threshing device, cleaning mechanism, seed pneumatic conveying device, elevator, seed bin, sunflower tray box, etc. The seed pneumatic conveying device includes a fan and pneumatic conveying pipe, as shown in Figure 1.

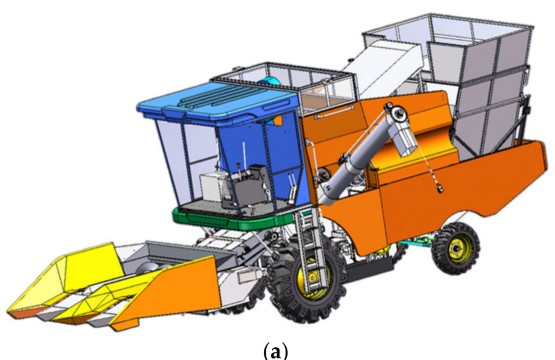　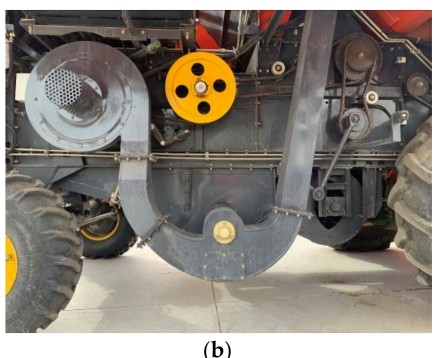

(**a**)　　　　　　　　　　　　　　　　　　　　　　　　(**b**)

**Figure 1.** 4ZXRKS-4 type self-propelled sunflower combine harvester: (**a**) sunflower combine harvester structure sketch; and (**b**) pneumatic conveying device.

The self-designed impact-type sunflower yield sensor is installed at the outlet of the seed pneumatic conveying pipe from this combine harvester, as shown in Figure 2.

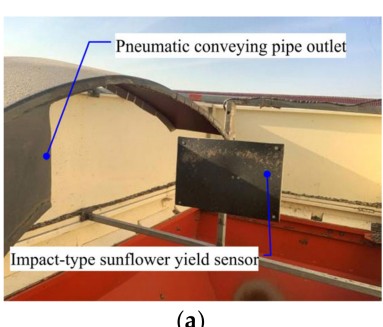
(a)

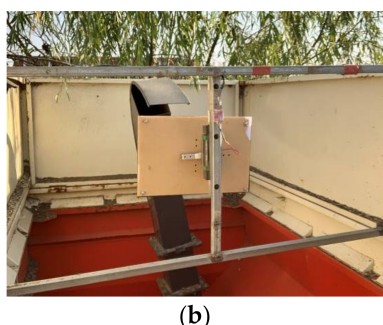
(b)

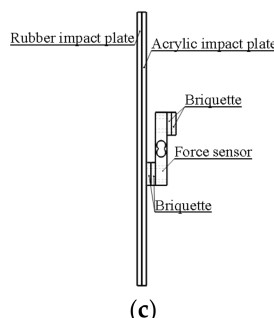
(c)

**Figure 2.** Impact-type sunflower yield sensor: (**a**) Front view; (**b**) Rear view; (**c**) Structure sketch.

The impact-type sunflower yield sensor consists of two major parts, namely the force sensor and the impact plate, and both are rigidly fixed as one. Among them, the force sensor uses an HL-8 double-hole cantilever parallel beam strain gauge force sensor, whose structure is mainly divided into three parts: elastomer, resistance strain gauge group bridge, and the power supply circuit of the bridge. In the sunflower harvesting process, sunflower seeds are threshed, sieved, and cleaned under the action of centrifugal fans, and continuously impacted on the impact plate of the sensor along the pneumatic conveying pipeline along the seed outlet. In order to reduce the mechanical damage caused to the sunflower seeds in the process of measuring the yield, the impact plate is composed of a rubber buffer plate and acrylic plate stacked back and forth, while using the pressure block to adjust the distance between the impact plate, the force sensor, and the mounting bracket, so as to avoid the measurement error caused by the phenomenon of rigid contact. This causes the deformation of the elastic element affixed with resistance strain gauge, and this deformation is converted into a voltage signal by the resistance strain gauge bridge, and this signal is passed through the differential amplifier circuit to obtain the voltage value of the measurement circuit, but noise, such as the vibration of the combined harvester, will affect the accuracy of the measurement results and increase the complexity of data analysis.

*2.2. Sunflower Yield Signal Characteristics*

In this study, the output signal of the impact type sunflower seed yield sensor from the 2021 field trial was selected for analysis. The voltage signal of the sensor was collected by STC12C5A60S2 micro controller and saved in real time by using the serial assistant. Figure 3 shows the change curve of the voltage signal of the impulse sunflower seed flow sensor in a certain vehicle in the yield measurement test, and the sampling frequency is set to 50 Hz. Section I is the sensor installation when it is ready and the signal acquisition starts; section II is the harvester start and fan start; section III is the fan start having finished and in a stable state; and section IV is the real-time harvesting process of the harvester. The signal was mainly distributed below 400 mV, with an amplitude range 0~818 mV, a mean value of 252.39 mV, a standard deviation of 173.26 mV and a variation coefficient of 69%.

In order to reduce the cost of microcontroller computing, this paper only analyzed and de-noised the sensor output signals of the segment IV real-time harvesting process. The autocorrelation function and partial correlation function were generated in EViews 9.0 software (Developed by Quantitative Micro Software) after zero-meaning the sampled signals from any two segments of yield measurement, as shown in Figure 4. If the auto correlation and partial correlation functions of the sampled signals are neither truncated nor trailed, the output signal of the impact-type sunflower yield sensors are determined to be a non-stationary random signal.

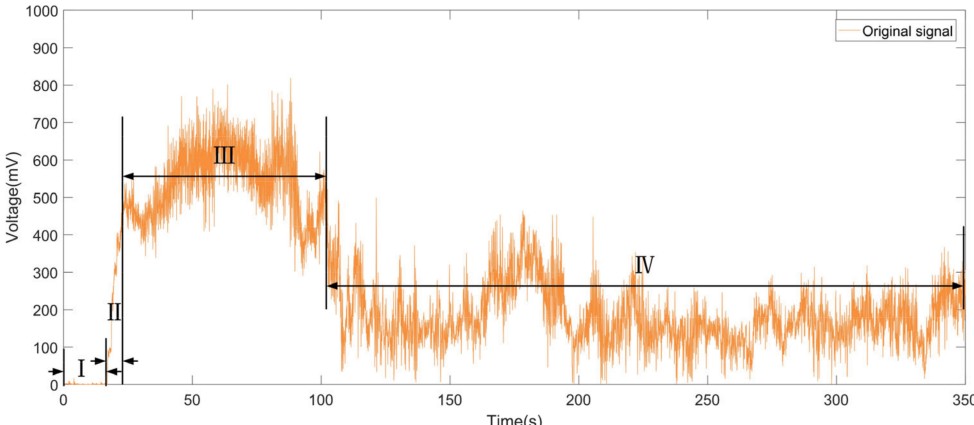

**Figure 3.** Impact-type sunflower yield sensor voltage original signal.

| Autocorrelation | Partial Correlation | | AC | PAC | Q-Stat | Prob |
|---|---|---|---|---|---|---|
| | | 1 | -0.036 | -0.036 | 0.1799 | 0.671 |
| | | 2 | -0.031 | -0.033 | 0.3186 | 0.853 |
| | | 3 | 0.002 | -0.000 | 0.3193 | 0.956 |
| | | 4 | -0.090 | -0.091 | 1.4909 | 0.828 |
| | | 5 | -0.022 | -0.029 | 1.5627 | 0.906 |
| | | 6 | -0.028 | -0.037 | 1.6785 | 0.947 |
| | | 7 | -0.013 | -0.017 | 1.7019 | 0.974 |
| | | 8 | -0.024 | -0.037 | 1.7903 | 0.987 |
| | | 9 | 0.146 | 0.140 | 5.0022 | 0.834 |
| | | 10 | 0.386 | 0.402 | 27.506 | 0.002 |
| | | 11 | -0.075 | -0.023 | 28.366 | 0.003 |
| | | 12 | 0.014 | 0.028 | 28.394 | 0.005 |

| Autocorrelation | Partial Correlation | | AC | PAC | Q-Stat | Prob |
|---|---|---|---|---|---|---|
| | | 1 | 0.033 | 0.033 | 0.1504 | 0.698 |
| | | 2 | -0.031 | -0.032 | 0.2839 | 0.868 |
| | | 3 | 0.041 | 0.043 | 0.5229 | 0.914 |
| | | 4 | -0.042 | -0.046 | 0.7794 | 0.941 |
| | | 5 | -0.046 | -0.040 | 1.0829 | 0.956 |
| | | 6 | 0.186 | 0.186 | 6.1323 | 0.409 |
| | | 7 | 0.015 | 0.001 | 6.1639 | 0.521 |
| | | 8 | 0.025 | 0.039 | 6.2603 | 0.618 |
| | | 9 | -0.005 | -0.026 | 6.2636 | 0.713 |
| | | 10 | 0.025 | 0.042 | 6.3557 | 0.785 |
| | | 11 | 0.009 | 0.021 | 6.3667 | 0.848 |
| | | 12 | -0.026 | -0.058 | 6.4694 | 0.891 |

(**a**)                    (**b**)

**Figure 4.** Sampled signal autocorrelation function and partial correlation function: (**a**) Sampling signal 1; and (**b**) Sampling signal 2.

*2.3. Algorithm Theory*

2.3.1. Complementary Ensemble Empirical Mode Decomposition (CEEMD)

CEEMD is a noise-assisted analysis method with the advantages of EMD to deal with non-stationary signals, and the added auxiliary noise is in the form of positive and negative pairs, which can effectively solve the phenomenon of EMD modal mixing and the error of reconstructing the model due to EEMD white noise residue [23]. This method is computationally efficient and is an effective algorithm for dealing with nonlinear and non-stationary signals [24,25].

The main steps of the CEEMD method are shown below.

Step 1: Add white noise of positive and negative paired form with amplitude $k$ and $I$ times to the original noisy signal to generate an ensemble signal, where $I$ is the average number of the ensemble. Generally, $I$ is taken as 100 and $k$ is taken as 0.02 [26];

Step 2: Apply EMD to decompose each ensemble signal to obtain $2I$ sets of intrinsic mode functions (IMF) components;

Step 3: The $j$-th IMF component of the signal decomposition is obtained by summing and averaging the $2I$ sets of IMF components, denoted as

$$c_j = \frac{1}{2I} \sum_{i=1}^{2I} c_{ij} \qquad (1)$$

where $c_j$ denotes the $j$-th IMF component of the CEEMD decomposition and $c_{ij}$ denotes the $j$-th IMF component of the $i$-th signal.

2.3.2. Wavelet Threshold Denoising (WTD)

The basic principle of WTD is to set a critical threshold $\lambda$. If the wavelet coefficients $< \lambda$, the coefficients are mainly generated by noise and this part of the coefficients is removed; if

the wavelet coefficients > λ, the coefficients are mainly generated by the signal and this part of the coefficients is retained, and finally the processed wavelet coefficients are inverted to obtain the de-noised signal [27,28]. It is essentially a band-pass filter, which acts as a series of low-pass and high-pass filters equivalent [29–31].

The main steps of WTD are shown below.

Step 1: Do N-point discrete sampling of $F(t)$ to get the discrete signal $F(n)$, $n = 0, 1, \ldots$ , $N−1$, after a suitable wavelet basis and the best wavelet decomposition layer $j$, the wavelet transform is

$$\omega_{j,k} = 2^{-j/2} \sum_{n=1}^{N-1} F(n)\psi(2^{-j}n - k) \tag{2}$$

A set of wavelet decomposition coefficients is obtained $\omega_{j,k}$, where $j$ is the number of wavelet decomposition layers, i.e., the wavelet decomposition scale, and $k$ denotes the position. $\omega_{j,k}$ consists of two parts, one is the wavelet coefficients $u_{j,k}$ corresponding to the useful signal, and the other is the wavelet coefficients $v_{j,k}$ corresponding to the noise.

Step 2: For the high-frequency coefficients of each layer from layer 1 to layer j, a reasonable threshold function is selected to correct the wavelet coefficients, and the estimated wavelet coefficients $\hat{\omega}_{j,k}$ are obtained by implementing a threshold quantization process for the high-frequency coefficients, and making $\left\|\hat{\omega}_{j,k} - u_{j,k}\right\|$ as small as possible.

Step 3: The estimated wavelet coefficients $\hat{\omega}_{j,k}$ after the threshold quantization process are wavelet inverse transformed to complete the signal reconstruction and obtain the estimated signal $\hat{F}(n)$, which is the signal after wavelet threshold denoising.

Therefore, the wavelet basis function, the number of decomposition layers, the threshold function and the threshold value need to be determined when using WTD for the analysis of sensor signals. Only by choosing appropriate parameter values for these key parameters can the desired noise reduction effect be achieved [32].

### 2.3.3. WTD Denoising Method Based on CEEMD

For the sunflower yield measurement signal in this study, when the noisy yield measurement signal contains valid information about the smaller amplitude, direct wavelet threshold denoising will remove most of the noise while removing some high-frequency valid signals [33]. In contrast, direct discarding of high-frequency IMF components when using CEEMD decomposition denoising will cause the problem of incomplete removal of high-frequency noise or loss of high-frequency effective signals. Meanwhile, it is difficult to choose the best parameters for the wavelet basis and threshold of the wavelet threshold denoising method, and the inappropriate wavelet basis and threshold directly affect the quality of denoising, but it can suppress the effect of modal aliasing and facilitate the noise reduction of IMFs with different noise contents; the CEEMD method has strong application adaptability, which can make up for the shortcomings of the length of the wavelet basis function and the decomposition transform method cannot be adaptive [34]; considering the non-smoothness of the impulse sunflower seed flow sensor signal and the adaptability of the denoising method to the non-smooth signal, this paper combined CEEMD with the wavelet threshold denoising method to solve the above problems in the denoising process by a complementary approach.

In this paper, we took advantage of CEEMD to decompose the vibration signal into several IMF. The noise component was then removed by calculating the correlation coefficient of the IMF. Since there was still noise in the removed noise signal, this noise was denoised by the wavelet threshold denoising method. Finally, the processed signal was reconstructed to obtain the final denoised signal. Figure 5 shows the flow chart of WTD based on CEEMD.

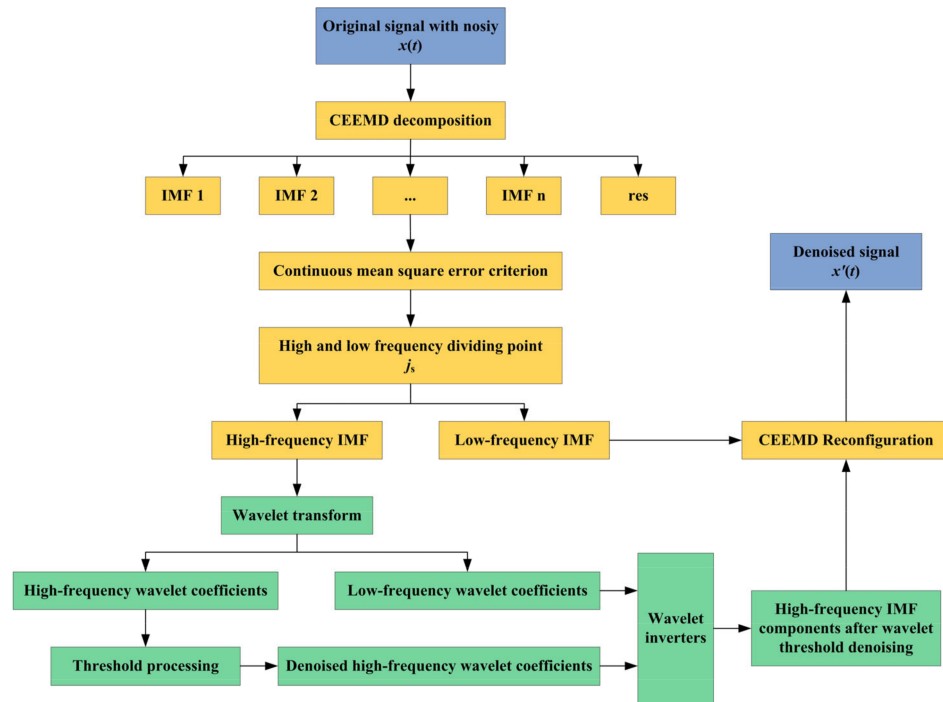

**Figure 5.** Impact-type sunflower yield sensor signal denoising flow chart combining CEEMD and WTD.

Step 1: The original signal $x(t)$ is CEEMD-decomposed to obtain each IMF component, which satisfies a series distribution from high to low frequencies, with the first few components being high-frequency components and random noise distributed in these high-frequency IMF components.

Suppose the reconfiguration signal $\widetilde{y}_k$ is

$$\widetilde{y}_k = \sum_{j=k}^{n} \mathrm{IMF}_j(t) + r_n(t), k = 1, 2, \ldots, n-1 \tag{3}$$

where: $\mathrm{IMF}_j(t)$ is the $j$-th order IMF component obtained by CEEMD decomposition; $n$ is the total number of IMF components; $r_n(t)$ is the residual component.

Step 2: According to the criterion of calculating the continuous mean square error, the boundary between the signal and the noise (i.e., the high-frequency and low-frequency IMF demarcation points) is determined, resulting in

$$\mathrm{CMSE}(\widetilde{y}_k, \widetilde{y}_{k+1}) = \frac{1}{N}\sum_{i=1}^{N}\left[\widetilde{y}_k(t_i) - \widetilde{y}_{k+1}(t_i)\right]^2 = \frac{1}{N}\sum_{i=1}^{N}\left[\mathrm{IMF}_k(t_i)\right]^2 \tag{4}$$

where: $N$ is the signal length; $[\mathrm{IMF}_k(t_i)]$ is the $k$-th order IMF component after CEEMD decomposition.

The CMSE calculated for this signal is taken as the minimum value to obtain the high and low frequency cutoffs $j_s$:

$$j_s = \underset{1 \leq k \leq n-1}{\mathrm{argmin}}\left[\mathrm{CMSE}(\widetilde{y}_k, \widetilde{y}_{k+1})\right] \tag{5}$$

By calculating the minimum value of the continuous mean square error of the measured output signal, the demarcation point $j_s = m$ between the signal and the noise is obtained, and the wavelet threshold denoising is done for each high frequency IMF before this point to obtain the first $m$ IMF components, i.e., $\mathrm{IMF}_j'$ (j = 1, 2, ... , $m$).

Step 3: The high frequency IMF component $\text{IMF}_j{}'$, the remaining low frequency component $\text{IMF}_j{}''$ and the residual $r_n(t)$ after wavelet threshold denoising are reconstructed to finally obtain the sunflower seed yield measurement signal after denoising, as

$$x'(t) = \sum_{j=1}^{m} \text{IMF}_j{}'(t) + \sum_{j=m+1}^{n} \text{IMF}_j{}''(t) + r_n(t) \tag{6}$$

Among them, the key parameters of WTD are specifically set as follows.

(1) Wavelet basis function.

Wavelet basis functions are commonly Bior wavelet, dB wavelet, Sym wavelet and Ciof wavelet. To determine the noise reduction level of the four wavelet bases, Bior, dB, Sym and Ciof are used as the test signals, and the noise standard deviation σ = 2 is set for signal decomposition, and the signal pseudo-noise reduction is evaluated by the signal-to-noise ratio and root mean square error as the signal pseudo-noise reduction criterion. According to the denoising effect, Sym8 signal-to-noise ratio is the largest, 5.6651; root mean square error is the smallest, 4.0126. Therefore, Sym8 wavelet basis is chosen in this paper.

(2) Number of decomposition layers.

The number of wavelet decomposition layers determines the separation effect of noise and signal. In order to achieve a good denoising effect and signal reconstruction without distortion, this paper compared the noise reduction effect of different decomposition layers of wavelet threshold denoising of measured signal and found that the best noise reduction effect is achieved when the number of decomposition layers of wavelet threshold denoising of measured signal is four layers.

(3) Threshold function.

Currently, the commonly used threshold functions are mainly hard threshold functions and soft threshold functions [35].

Hard threshold function:

$$\hat{W}_{j,k} = \begin{cases} W_{j,k} & \left| W_{j,k} \right| \geq \lambda \\ 0 & \left| W_{j,k} \right| < \lambda \end{cases} \tag{7}$$

where, $\hat{W}_{j,k}$ denotes the wavelet coefficients after threshold denoising; $W_{j,k}$ denotes the wavelet decomposition coefficients; and λ denotes the threshold value.

Soft threshold function:

$$\hat{W}_{j,k} = \begin{cases} \text{sign}(W_{j,k})(\left| W_{j,k} \right| - \lambda) & \left| W_{j,k} \right| > \lambda \\ 0 & \left| W_{j,k} \right| < \lambda \end{cases} \tag{8}$$

where $\hat{W}_{j,k}$ is the wavelet coefficient after threshold denoising; $W_{j,k}$ is the wavelet decomposition coefficient; $sign(\cdot)$ is the sign function; λ is the threshold value; N is the total length of the signal; and σ is the standard deviation of the noise.

To verify the denoising effects of hard and soft threshold functions, different amounts of white noise are added to the original data, and the Sym8 wavelet basis is selected to do 4-layer smooth wavelet decomposition of the noise-containing signal under different signal-to-noise conditions, and the high-frequency coefficients on each time scale are denoised by applying hard and soft threshold functions, respectively. The results show that the signal-to-noise ratio after soft threshold is significantly higher than that of hard threshold under the same condition of signal-to-noise ratio. Therefore, the soft threshold function is chosen in this paper.

(4) Threshold

When performing wavelet threshold denoising on a signal, the size of the threshold plays a decisive role in the denoising effect. There are four main methods commonly used for threshold selection: unbiased likelihood estimation threshold, very large and very small threshold, fixed threshold and heuristic threshold criterion [36], among which,

heuristic threshold is a combination of fixed threshold criterion and unbiased likelihood estimation threshold criterion, which is a more reasonable selection criterion for predicting the threshold of variables. Therefore, the heuristic threshold criterion is chosen in this paper.

Let $s$ be the sum of squares of $N$ wavelet decomposition coefficients such that $\eta = (s - N)/N, \mu = (\log_2 N)^{2/3} \sqrt{N}$. Then, the heuristic threshold $\lambda^h$ is calculated as shown:

$$\lambda^h = \begin{cases} \lambda^s & \eta \leq \mu \\ \min(\lambda^s, \lambda^r) & \eta > \mu \end{cases} \tag{9}$$

where $\lambda^s$ is the fixed threshold and $\lambda^r$ is the unbiased likelihood estimation threshold, calculated from Equations (9) and (10), respectively.

$$\lambda^s = \delta \sqrt{2 \log N} \tag{10}$$

$$\begin{cases} Rigrsure(\lambda) = \frac{\|w_\lambda - w\|^2 + (N - 2N_0)\delta^2}{N} \\ \lambda^r = \underset{0 \leq \lambda \leq \sqrt{2 \log N}}{\arg\min} \ Rigrsure(\lambda) \end{cases} \tag{11}$$

where $\delta$ is the noise standard deviation; $N$ is the signal length; $w_\lambda$ is the wavelet coefficients obtained using threshold shrinkage; $w$ denotes the original wavelet coefficients; $N_0$ is the number of wavelet coefficients after they are set to zero.

2.3.4. Criteria for Judging Denoising Effect

Based on the requirements of sunflower yield measurement signals, this study constructed an effect judgment criterion applicable to the signal denoising of impact-type sunflower yield sensor. According to the signal-to-noise ratio (SNR), root mean square error (RMSE), smoothness (S) and waveform similarity (WS) of the denoised signal, where the larger the SNR and WS the better, and the smaller the RMSE and S the better [37,38]. Considering the case where RMSE and S are equal to zero, the reciprocal of (1 + RMSE) is called the algorithmic approximation and is denoted as $\text{RMSE}^{-1}$, and obviously, $0 < \text{RMSE}^{-1} \leq 1$; the reciprocal of (1 + S) is called the algorithmic smoothness and is denoted as $\text{S}^{-1}$, then $0 < \text{S}^{-1} \leq 1$.

In this paper, different weights were given to the above indicators according to the requirements of the measured output signal, and the following criteria were established to determine the denoising effect.

Constraints:

$$\begin{cases} \max\{\text{SNR}\} = \max\left\{ 10 \lg\left( \sum\limits_{i=1}^{n} x_i^2 \Big/ \sum\limits_{i=1}^{n} \left(x_i - \breve{x}_i\right)^2 \right) \right\} \\ \max\left\{\text{RMSE}^{-1}\right\} = \max\left\{ 1 \Big/ 1 + \sqrt{\frac{1}{n} \sum\limits_{i=1}^{n} \left(x_i - \breve{x}_i\right)^2} \right\} \\ \max\left\{\text{S}^{-1}\right\} = \max\left\{ 1 \Big/ \left( 1 + \sum\limits_{i=1}^{n-1} \left[\breve{x}_{i+1} - \breve{x}_i\right]^2 \Big/ \sum\limits_{i=1}^{n-1} [x_{i+1} - x_i]^2 \right) \right\} \\ \max\{\text{WS}\} = \max\left\{ \sum\limits_{i=1}^{n} \breve{x}_i x_i \Big/ \sqrt{\sum\limits_{i=1}^{n} x_i^2 \sum\limits_{i=1}^{n} \breve{x}_i^2} \right\} \end{cases} \tag{12}$$

where $x_i$ is the original signal; $\breve{x}_i$ is the denoised signal; $n$ is the signal length.

The denoising algorithm evaluates the objective function $f$ as

$$\max\{f\} = \max\left\{ \alpha\text{SNR} + \beta\text{RMSE}^{-1} + \gamma\text{S}^{-1} + \delta\text{WS} \right\} \tag{13}$$

where $\alpha$ is the influence factor of SNR; $\beta$ is the influence factor of $\text{RMSE}^{-1}$; $\gamma$ is the influence factor of $\text{S}^{-1}$; $\delta$ is the influence factor of WS; and $\alpha + \beta + \gamma + \delta = 1$.

Consider that the signal post-processing work is mainly for yield measurement signal extraction and yield analysis. Therefore, the smaller the noise in the signal is expected to be better. We pay more attention to the denoising ability and denoising effect of the denoising method, so we set the SNR and RMSE$^{-1}$ influence factor to be larger, at 0.3. The influence factor of S$^{-1}$ and WS is set to be smaller, at 0.2, i.e., $\alpha = \beta = 0.3$ and $\gamma = \delta = 0.2$.

## 3. Results and Discussion

### 3.1. Simulation Signal Verification

Considering the actual working conditions of the sunflower combine harvester, the impact-type sunflower yield sensor was approximated as a single-degree-of-freedom second-order linear system in this paper. Meanwhile, the simulated output signal of the impact-type sunflower yield sensor was constructed by combining the impact function of single grain [39]. Gaussian white noise was added to the simulated signal as the noise in the measured yield. The simulated signal was set as

$$x(t) = s(t) + n(t) \tag{14}$$

$$s(t) = \frac{\hat{F}}{m\sqrt{1 - \xi^2}} e^{-\xi \omega_n t} \sin\left(\sqrt{1 - \xi^2} \omega_n t\right) \tag{15}$$

where $x(t)$ is the noise-containing simulation signal output from the sensor, $s(t)$ is the pure simulation signal, $n(t)$ is the Gaussian white noise signal, $\hat{F}$ is the pulse force, taking the range of the sensor as 5 N; $\omega_n$ is the intrinsic frequency of the sensor, taking 28.105 Hz; $\zeta$ is the damping ratio of the sensor, taking 0.7; $m$ is the mass of a single sunflower seed, taking 0.0002 kg; the Gaussian white noise amplitude is set to 0.5, and the sampling frequency is 50 Hz.

The time domain signals of the pure simulation signal and the noisy simulation signal are shown in Figure 6.

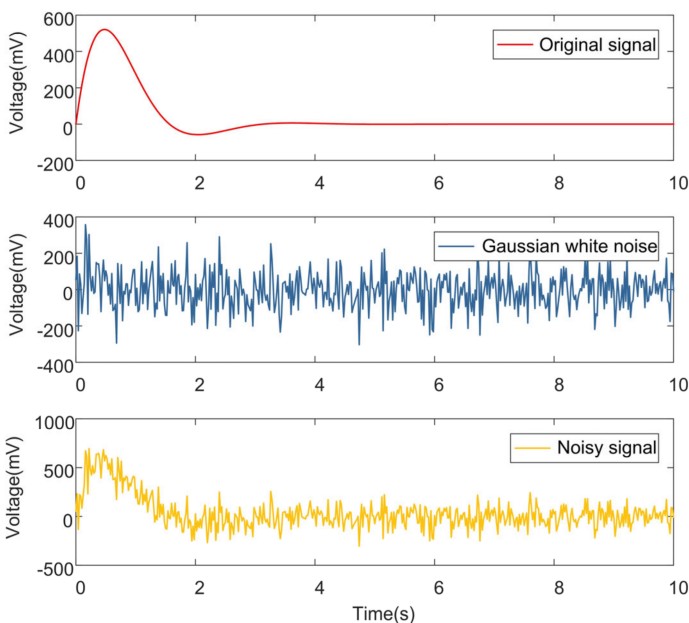

**Figure 6.** Time domain signal diagram of the simulated signal.

To compare the effectiveness of existing decomposition algorithms for denoising, this paper performs EMD, EEMD and CEEMD decomposition on the numerically simulated impact-type sunflower yield sensor output signal x(t). Among them, the white noise signal-to-noise ratio in the EEMD and CEEMD algorithms is set to 0.2, the overall average number of iterations is 100, and the maximum number of iterations is 1000. the decomposition results are shown in Figure 7. As can be seen from Figure 7, EMD adaptively decomposes

the simulated signal into 8 components, i.e., IMF1~IMF7 and residual res; EEMD and CEEMD adaptively decompose the simulated signal into 10 components, i.e., IMF1~IMF9 and residual res. The IMF components obtained after the decomposition of EMD and EEMD have obvious modal mixing problems: IMF1 does not extract the high-frequency components separately but adulterates the low-frequency ones. Similarly, the other IMFs contain fluctuating factors of different frequencies. This makes the IMF lose the characteristics of a single feature scale, complicating the signal analysis and making it impossible to identify the exact components in each IMF. In contrast, the IMF components obtained after CEEMD decomposition are least affected by modal confounding, and the obtained IMF5 contains only one time-scale feature component, which is beneficial to the subsequent signal denoising.

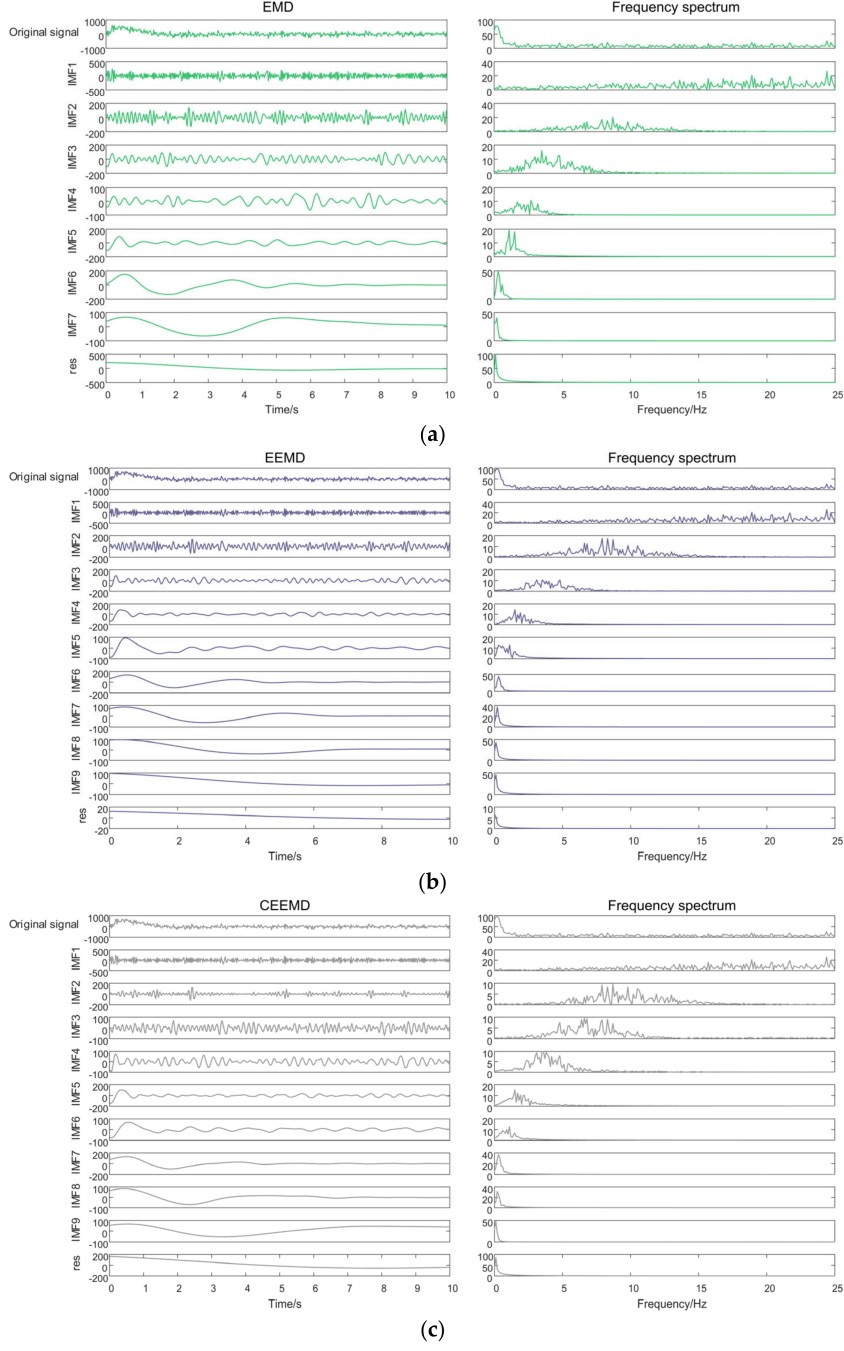

**Figure 7.** Simulation signal decomposition: (**a**) EMD; (**b**) EEMD; and (**c**) CEEMD.

The energy distribution of each order IMF component is obtained according to the continuous mean square error calculation Equation (3) as shown in Figure 8, and it is determined that the IMF of local energy abrupt change is the demarcation point of noise and signal. Therefore, the demarcation point $j_s$ = 4 for EMD, $j_s$ = 4 for EEMD and $j_s$ = 4 for CEEMD in this paper.

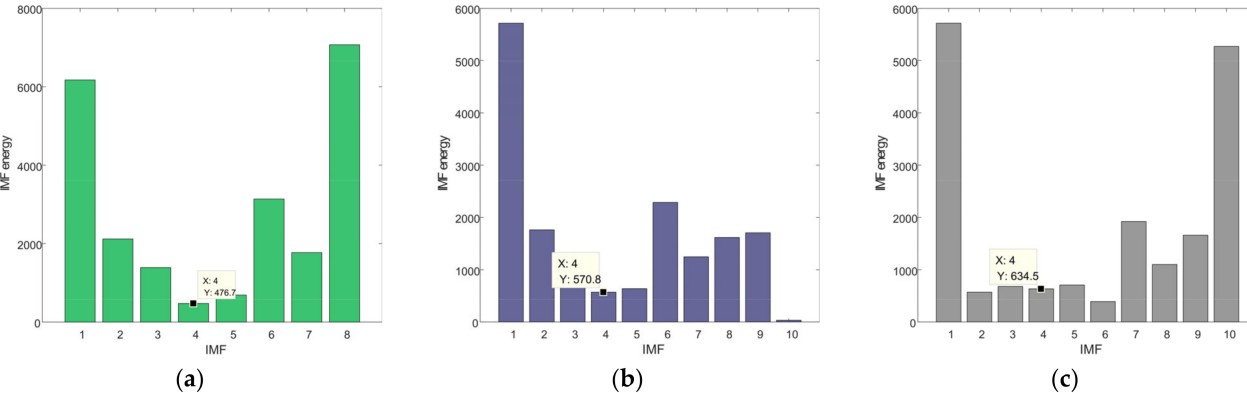

**Figure 8.** IMF energy per order for the simulated signal: (**a**) EMD; (**b**) EEMD; (**c**) CEEMD.

According to the demarcation point $j_s$ of EMD, EEMD and CEEMD, the IMF components after the 5-th order of EMD, EEMD and CEEMD are reconstructed separately, and the simulated signals denoised by EMD, EEMD and CEEMD decomposition are obtained as shown in Figure 9. It can be seen that the noise reduction effect of EMD, EEMD and CEEMD is basically ideal, which can effectively suppress the noise, and the denoised signal is smooth and almost burr-free. the waveform after denoising of CEEMD can better restore the fluctuation of the simulated signal compared with EMD and EEMD, and EEMD appears to be smoother. However, since the high frequency component still contains valid information, the high frequency component is directly discarded according to the high and low frequency dividing points, which makes the signal amplitude decrease while noise reduction.

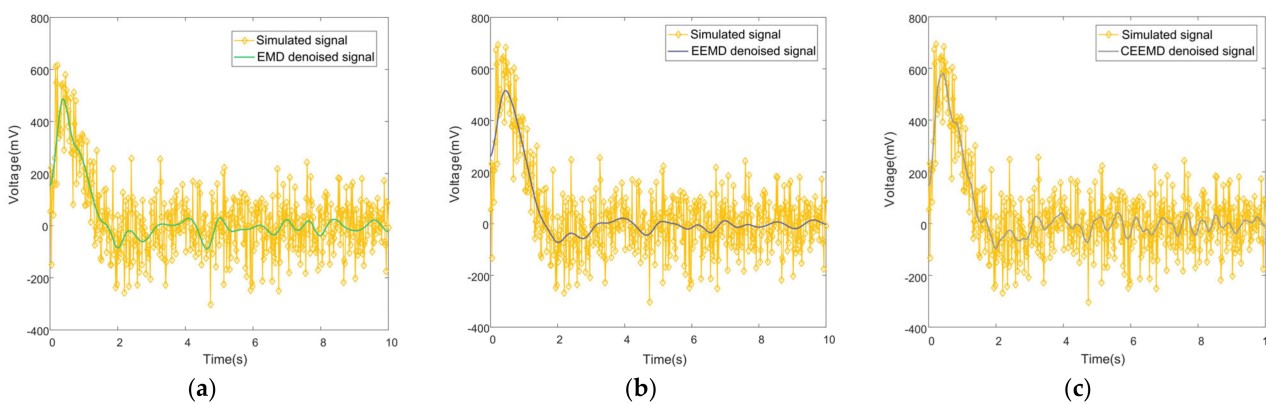

**Figure 9.** Time domain diagram of the simulated signal before and after denoising: (**a**) EMD; (**b**) EEMD; (**c**) CEEMD.

Meanwhile, the simulated signal is processed by using wavelet threshold denoising method, and according to the parameters determined above, the wavelet basis function is chosen as Sym8, the number of decomposition layers is 4, and the results of wavelet multilayer decomposition are shown in Figure 10. Where, a is the approximation coefficient and d is the detail coefficient. Therefore, the low-frequency signal corresponds to the approximate signal a4 of the decomposition, which has the same trend as the simulated signal; the high-frequency signal corresponds to the detail signal d1 of the decomposition, which is mainly the noise signal. A soft threshold function with a threshold selection

heuristic threshold criterion is used to perform wavelet threshold denoising on the high-frequency part obtained from wavelet decomposition and reconstruct it with the low-frequency signal, and the results are shown in Figure 11. It can be found that the simulated signal after wavelet threshold denoising can filter out the high-frequency noise well, and the characteristics of the original signal can be well retained.

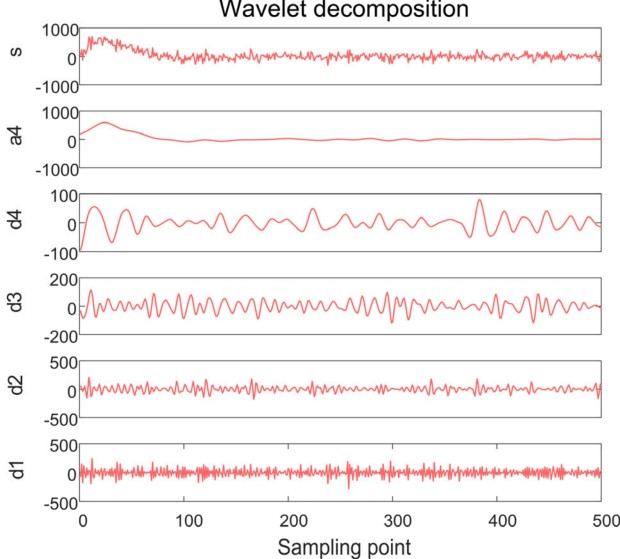

**Figure 10.** Simulated signal wavelet decomposition.

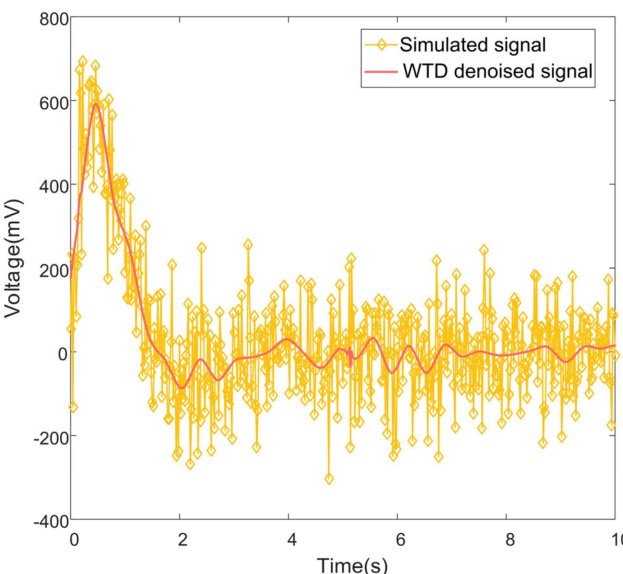

**Figure 11.** Simulated signal before and after wavelet threshold denoising.

In addition, the simulated signal is denoised using the method CEEMD-WTD proposed in this paper, and the denoising steps are shown in 2.3.3. The denoising results of CEEMD-WTD are shown in Figure 12, and the denoising effect is relatively satisfactory, and the method effectively avoids the weakening of signal features after noise reduction caused by the direct removal of high-frequency components.

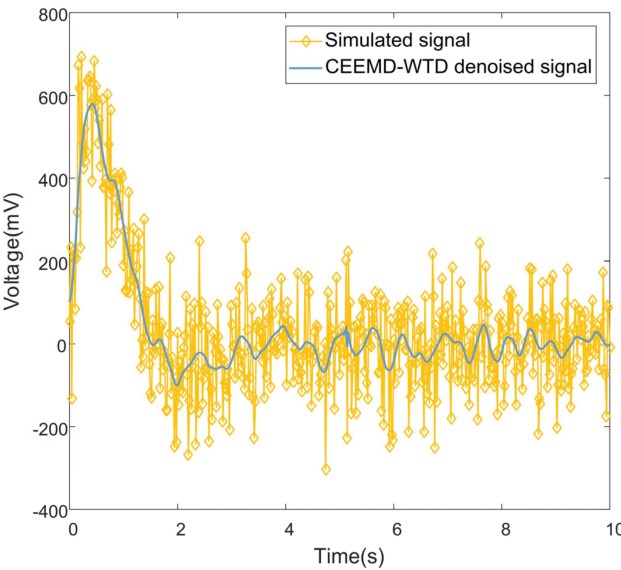

**Figure 12.** Simulated signal before and after CEEMD-WTD denoising.

In order to compare and analyze the advantages of the proposed method CEEMD-WTD denoising method and other decomposition denoising methods (EMD, EEMD, CEEMD, WTD), the denoising results of the above five denoising methods on the simulated signals are compared and analyzed, and the denoising results are shown in Figure 13.

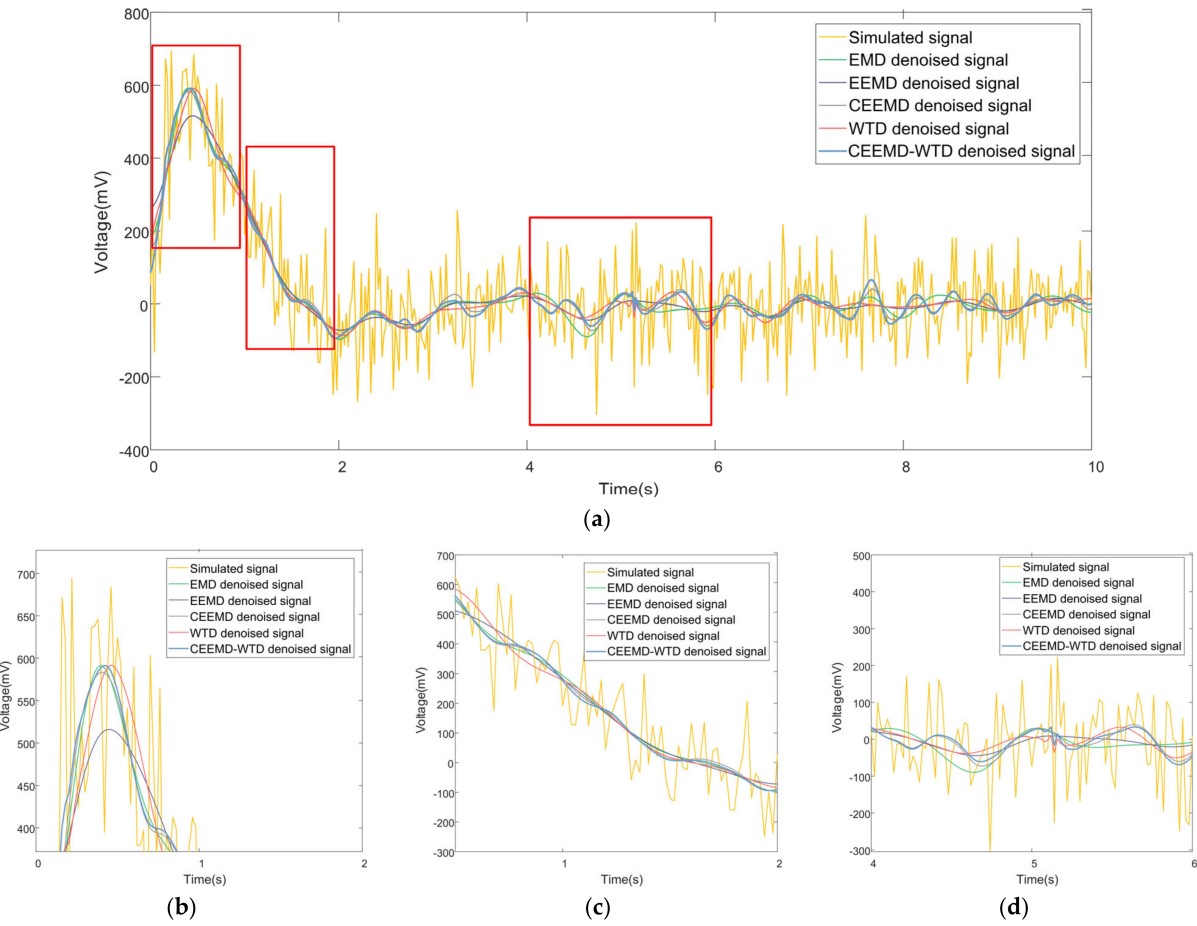

**Figure 13.** Time domain diagram of the simulated signal after denoising: (**a**) global signal; (**b**) locally amplified signals-1; (**c**) locally amplified signals-2; and (**d**) locally amplified signals-3.

As can be seen in Figure 13, EMD, EEMD, CEEMD, WTD, and our newly proposed method (CEEMD-WTD) can all reduce the signal amplitude to reduce or eliminate the noise in the middle and high frequencies. The local amplified signals of 0–2 s, 0.5~2 s and 4~6 s are selected for clearer observation. Among them, EMD, EEMD, and CEEMD reduced the signal amplitude, while WTD showed some panning in the time axis, all of which caused the loss of useful information in the signal. The simulated signal after denoising based on the CEEMD-WTD method has a curve mostly between these five methods, and the data at the peaks and valleys are smoother, and the amplitude of the signal at the peaks and valleys is well preserved, which ensures the amplitude of the simulated signal to the maximum extent. Therefore, in general, the method proposed in this paper has a good denoising effect and is suitable for the denoising of the output signal of the impact-type sunflower yield sensor.

In order to quantitatively compare the effects of the above five noise reduction methods more intuitively from the data, we use SNR, RMSE, S, and WS as evaluation indexes, and combine the algorithm denoising effect judgment criterion constructed in this paper, i.e., the denoising algorithm objective function ($f$) as reference, to compare the simulated signals processed by the above noise reduction, and the comparison results are shown in Table 1. From Table 1, it can be seen that CEEMD-WTD has the largest SNR, WS and $f$ and the smallest RMSE compared with EMD, EEMD, CEEMD and WTD, then the CEEMD-WTD method proposed in this paper has the strongest denoising ability compared with the other four methods.

**Table 1.** Simulated signal denoising evaluation.

|  | SNR | RMSE | S | WS | $f$ |
|---|---|---|---|---|---|
| EMD | 3.8455 | 98.4998 | 0.0015 | 0.7665 | 1.5096 |
| EEMD | 4.8594 | 99.9524 | 0.0010 | 0.8209 | 1.8248 |
| CEEMD | 5.2715 | 95.3205 | 0.0031 | 0.8385 | 1.9516 |
| WTD | 5.0185 | 98.1376 | 0.0023 | 0.8277 | 1.8737 |
| CEEMD-WTD | 5.3375 | 94.5993 | 0.0037 | 0.8412 | 1.9719 |

### 3.2. Sensor Real Signal Verification

In this paper, the measured signals from a field impulse sunflower seed flow sensor were de-noised and analyzed on 16 October 2021, in Nai Zigai Township, Tokoto County, Hohhot City, Inner Mongolia Autonomous Region, China. The trial field was planted with an SH363 sunflower hybrid variety and the moisture content at harvest was 23.4%, which was within the standard moisture content range for sunflower harvests. The impulse sunflower seed mass flow sensor was equipped with a 4ZXRKS-4 self-propelled sunflower combine harvester (Inner Mongolia Hongchang Machinery Factory, China) with factory number XB0087. The field test is shown in Figure 14.

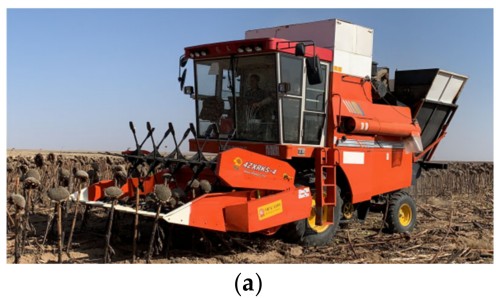
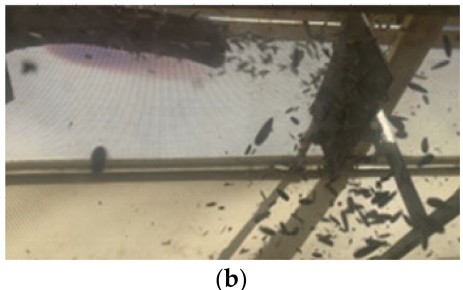

(**a**)                                                   (**b**)

**Figure 14.** Sunflower harvesting field experiment: (**a**) 4ZXRKS-4 type self-propelled sunflower combine harvester; and (**b**) impact-type sunflower yield sensor.

The output voltage signal of the impact-type sunflower yield sensor is shown in Figure 15.

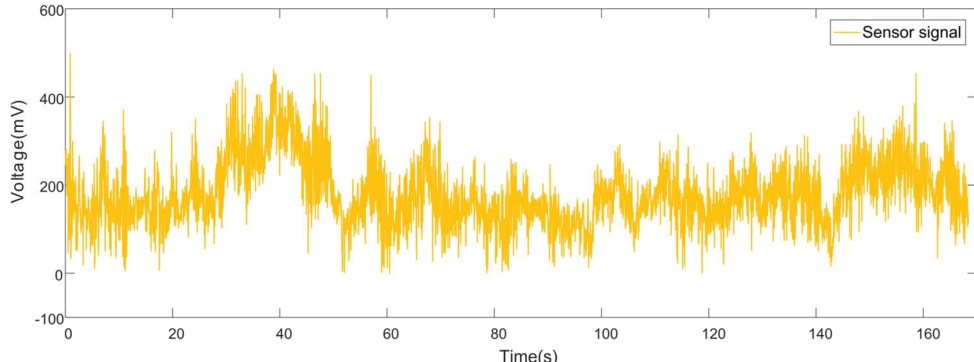

**Figure 15.** Sensor original signal.

In this paper, EMD, EEMD, CEEMD and wavelet decomposition were performed on the output signal of impact-type sunflower yield sensor. Among them, the white noise signal-to-noise ratio in EEMD and CEEMD algorithms was set to 0.2, the overall average number is 100, and the maximum number of iterations was 1000; the wavelet basis function of wavelet decomposition was selected as Sym8, and the number of decomposition layers was four. The decomposition results are shown in Figure 16. From the decomposition results, it can be seen that the frequency domain amplitude of the sensor output voltage signal is high, and the frequency is close to 0 Hz, which is mainly because the sensor signal is a DC component. the IMF components all meet the distribution from high frequency to low frequency, it can be seen that the high frequency noise mainly exists in the first three orders of IMF components, but there is still the phenomenon of modal aliasing. In the wavelet decomposition, the four-layer wavelet decomposition effectively reveals the high-frequency signal but cannot better realize the adaptive decomposition of the signal according to the time-scale characteristics of the sensor signal itself. EMD adaptively decomposes the sensor signal into 12 components, i.e., IMF1~IMF11 and residual res, and the high-frequency signal exists in IMF1~IMF5; EEMD adaptively decomposes the sensor signal into 13 components, i.e., IMF1~IMF12 and residual res, with high-frequency signals present in IMF1~IMF5; CEEMD adaptively decomposes the simulated signal into 14 components, i.e., IMF1~IMF13 and residual res, with high frequency signals present in IMF1~IMF7.

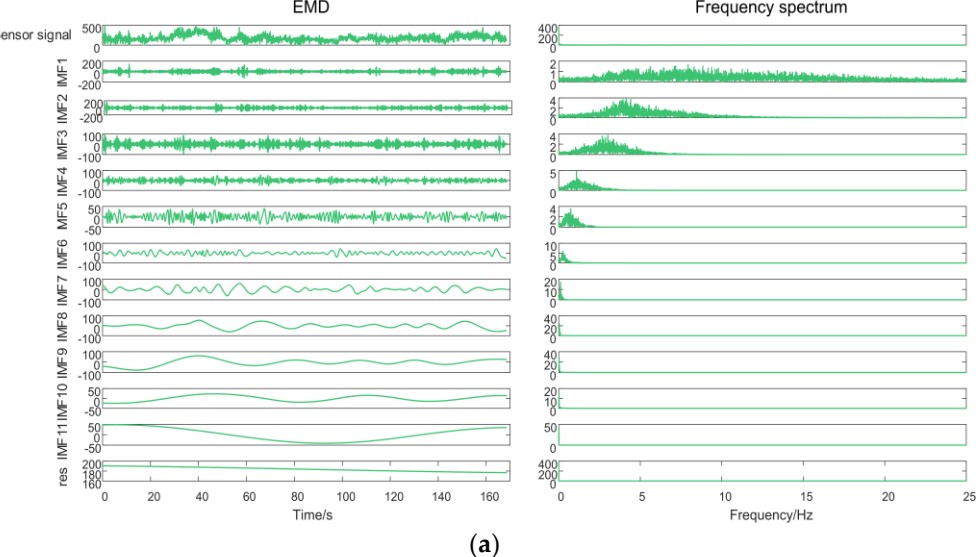

(**a**)

**Figure 16.** *Cont.*

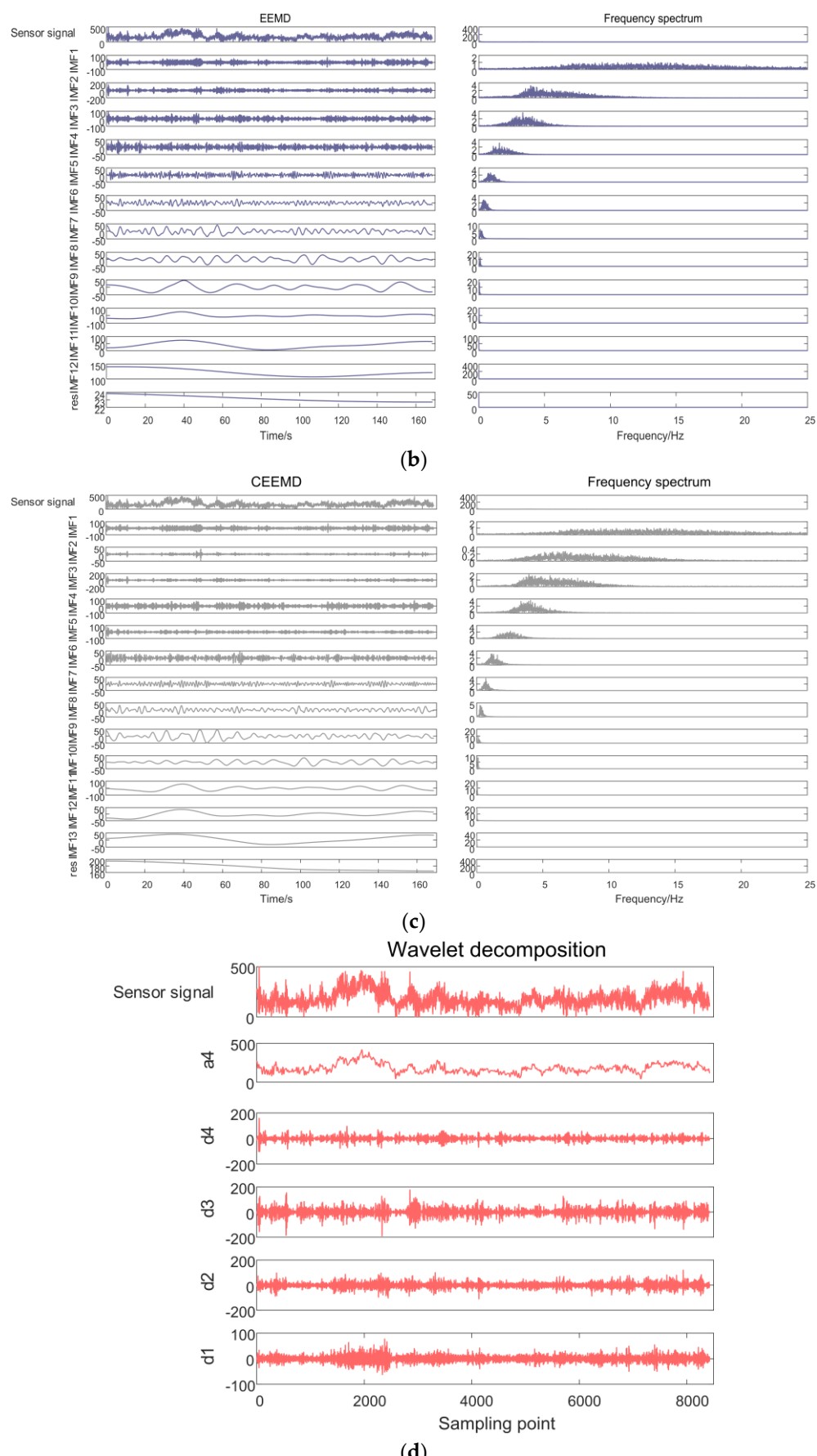

**Figure 16.** Sensor signal decomposition: (**a**) EMD; (**b**) EEMD; (**c**)CEEMD; and (**d**) wavelet.

To further determine the high and low frequency cut-off points accurately, the energy distribution of each order IMF component was obtained according to the continuous mean square error as shown in Table 2. It was finally determined that the cut-off points $j_s$ for EMD, EEMD and CEEMD of the sensor signals are all 5, and $j_s = 7$ for CEEMD.

**Table 2.** Sensor signal per order IMF energy.

| IMF | EMD | EEMD | CEEMD |
| --- | --- | --- | --- |
| 1 | 0.0394 | 0.0193 | 0.0193 |
| 2 | 0.0762 | 0.0498 | 0.0009 |
| 3 | 0.0433 | 0.0298 | 0.0226 |
| 4 | 0.0255 | 0.0126 | 0.0350 |
| 5 | 0.0171 | 0.0081 | 0.0128 |
| 6 | 0.0236 | 0.0103 | 0.0139 |
| 7 | 0.0456 | 0.0193 | 0.0085 |
| 8 | 0.0662 | 0.0223 | 0.0108 |
| 9 | 0.1039 | 0.0378 | 0.0270 |
| 10 | 0.0205 | 0.0432 | 0.0153 |
| 11 | 0.0920 | 0.1949 | 0.0653 |
| 12 | 3.3800 | 1.4921 | 0.0328 |
| 13 | - | 0.0537 | 0.0630 |
| 14 | - | - | 3.1563 |

Unit: $1 \times 10^4$.

The signals were denoised by EMD, EEMD, CEEMD, WTD and the method proposed in this paper, CEEMD-WTD, respectively. In the CEEMD-WTD method proposed in this paper, the CEEMD decomposition to obtain the first 7-th order IMF high-frequency components was processed by the wavelet threshold denoising method, and the processed IMF high-frequency components and the remaining IMF low-frequency components were reconstructed. Other decomposition methods use direct filtering of high-frequency components. Comparing the original signal, the EMD processed signal, the EEMD processed signal, the CEEMD processed signal, the WTD processed signal and the CEEMD-WTD processed signal proposed in this paper is shown in Figure 17, and the evaluation of the denoising effect is shown in Table 3.

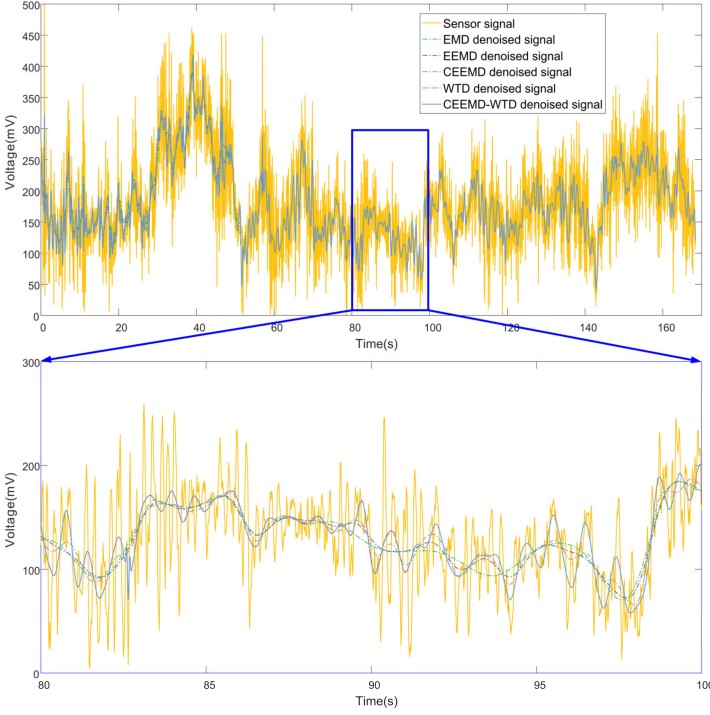

**Figure 17.** Time domain diagram of the sensor signal after denoising.

**Table 3.** Sensor signal denoising evaluation.

|  | SNR | RMSE | S | WS | $f$ |
|---|---|---|---|---|---|
| EMD | 12.6667 | 44.9673 | 0.00054 | 0.97257 | 4.0012 |
| EEMD | 12.9026 | 43.7625 | 0.00064 | 0.97404 | 4.0724 |
| CEEMD | 13.1225 | 42.6684 | 0.00136 | 0.97533 | 4.3384 |
| WTD | 13.7635 | 39.6329 | 0.01413 | 0.97876 | 4.5294 |
| CEEMD-WTD | 13.7715 | 39.5962 | 0.01412 | 0.97880 | 4.5318 |

It can be seen that EMD, EEMD, and CEEMD can provide reasonable decomposition results, and the signal smoothing effect is outstanding, which can obtain a better noise reduction effect, and the adaptive decomposition ability fundamentally gets rid of the interference of human factors. Among them, the SRN, RMSE, and WS of CEEMD are the highest, which indicates that the denoising and waveform restoration ability of CEEMD is stronger. Although waveform distortion exists in WTD, the SRN, RMSE, and WS are better than those of EMD, EEMD, and CEEMD, while the smoothing performance of EMD and EEMD is better, but the degree of detail retention is slightly worse. The CEEMD-WTD method proposed in this paper better preserves the main information of the impact-type sunflower yield sensor signal, suppresses the modal aliasing phenomenon of the adaptive decomposition process, and significantly reduces the burrs near the signal extremum point. Although the smoothing performance seems to be poor, the fluctuation of the signal is well restored, and the similarity between the denoised waveform and the original waveform is high, and the signal is almost undistorted, and the denoising effect is better than the other four algorithms both visually and quantitatively analyzed. It proves that the method proposed in this paper is applicable to the signal noise reduction of impact-type sunflower yield sensor with pneumatic grain delivery structure, which has good reasonableness and superiority.

## 4. Conclusions

(1) Analyzing the output signal of the impact-type sensor for sunflower combine harvester, it was observed that the autocorrelation function and partial correlation function of the sampled signal are neither truncated nor trailing, and the output signal of the impact-type sunflower yield sensor is determined to be a non-stationary random signal;

(2) Based on the characteristics of the non-smooth random signal of the impact-type sunflower yield sensor, a targeted CEEMD-WTD denoising method is proposed for the impact-type sensor signal adapted to the pneumatic seed delivery structure. The method combines the adaptive decomposition capability of CEEMD and the fast computation capability of WTD, which can meet the needs of field yield measurement random signal denoising and real-time yield measurement;

(3) By constructing the CEEMD-WTD algorithm denoising ability evaluation objective function ($f$) and evaluating the denoising effect with different weights of signal-to-noise ratio (SNR), root mean square error (RMSE), smoothness (S) and waveform similarity (WS), the algorithm achieves a balance between denoising effect and signal fidelity;

(4) By comparing and analyzing the denoising effect of the sunflower seed impact simulation signal and the actual measured signal of the field impulse type sunflower seed flow sensor, it can be seen that CEEMD-WTD has significant advantages over the existing single EMD, EEMD, CEEMD, and WTD methods in terms of SNR, RMSE, WS, and $f$, and is suitable for application to the pneumatic grain delivery structure of impact-type sunflower yield sensor signal de-noising.

**Author Contributions:** Conceptualization, S.W. and Z.Y.; methodology, S.W., X.Z. and Z.Y.; software, S.W. and W.L; validation, X.Z. and D.Z.; formal analysis, J.D.; resources, J.D.; data curation, W.L. and J.D.; writing—original draft preparation, X.Z. and D.Z.; writing—review and editing, S.W., D.Z. and Z.Y.; visualization, S.W. and W.L.; supervision, Z.Y.; project administration, Z.Y.; funding acquisition, Z.Y. All authors have read and agreed to the published version of the manuscript.

**Funding:** This research was funded by NATIONAL NATURAL SCIENCE FOUNDATION OF CHINA, grant number 51865047.

**Institutional Review Board Statement:** Not applicable.

**Informed Consent Statement:** Not applicable.

**Data Availability Statement:** Not applicable.

**Acknowledgments:** We would like to thank Changwei Niu, Aorigele, Yipeng Du, Zhizhen An, Jian Song, Xiaochao Chen, and Ke Wang for their vital support to the sunflower field experiment.

**Conflicts of Interest:** The authors declare no conflict of interest.

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
