# Peer review of "Impact-Type Sunflower Yield Sensor Signal Denoising Method Based on CEEMD-WTD"

_agriculture, doi:10.3390/agriculture13010166_

Round 1

Reviewer 1 Report

Please refer to the attached file to improve the article.

Reviewer 2 Report

This is a very good manuscript. The problem of signal noise in yield monitoring systems is valid and the authors have proposed a novel approach to address it. The manuscript is well written and organized, however some parts are suffering from poor English. I would suggest that the authors use a scientific tone. For example, in line 507: "we know that", could be removed, or replaced with terms such as "it was observed that".

I would also suggest adding more background on yield monitoring systems, perhaps by providing a figure in the introduction that shows one or two impact-type s yield sensors, load cells, etc in different yield monitoring systems for different crops. This would highlight the importance of this study. 

It is clear that the manuscript has been generated from an honest study. I liked the wavelet idea. It can be published as is, or after minor improvement of the English. Congrats to the authors, we need more quality manuscripts like this.

Reviewer 3 Report

1.In Introduction, the author should focus on the highlights of the proposed method and what problems have been solved. such as (1)(2)(3).

2.At present, there are many selection criteria for IMF signal components. Please explain why the continuous mean square error criterion (CMSE) is selected as the signal screening criterion in this paper.

3.The size of some mathematical formulas in this article is inconsistent, such as formula (13). Please standardize the typesetting of all the formulas.

4. Figure 16 in this paper is too small to see the signal decomposition result and spectrum diagram clearly. Please modify the Figure with similar situation. Meanwhile, The clarity of some figures in the text is too low, which makes it difficult to see the effects of different methods ,such as Figure 17.

Round 2

Reviewer 3 Report

None